# Absence of a Hernia Sack in Patients Undergoing Prenatal Repair of Spina Bifida Increases the Risk of Developing Shunt-Dependent Hydrocephalus

**DOI:** 10.3390/diagnostics13030343

**Published:** 2023-01-17

**Authors:** Agnieszka Pastuszka, Tomasz Koszutski, Ewa Horzelska, Sylwia Marciniak, Mateusz Zamłyński, Anita Olejek

**Affiliations:** 1Department of Gynecology, Obstetrics and Oncological Gynecology, Medical University of Silesia, 40-055 Katowice, Poland; 2Department of Pediatric Surgery and Urology, Faculty of Medical Sciences, Medical University of Silesia, 40-055 Katowice, Poland

**Keywords:** spina bifida, prenatal surgery, hydrocephalus

## Abstract

Spina bifida aperta (SBA), with (myelomeningocele) or without (myeloschisis) a hernia sack, is the most common congenital defect of the central nervous system. Prenatal surgical closure of SBA lowers the risk for developing shunt-dependent hydrocephalus, which offers a chance at improved motor, urinary, and gastrointestinal function. A total of 96 patients who had undergone open surgery prenatal repair for SBA were analyzed. The patients were divided into two groups: Group I—12 patients (12.5%)—without a hernia sack (myeloschisis) and Group II—84 patients (87.5%)—with a hernia sack (myelomeningocele). In this study, we demonstrated that prenatal SBA repair was statistically significantly less often associated with the need for ventriculoperitoneal shunting (*p* > 0.00001). The shunting was statistically significantly more often required in patients from Group I (*p* > 0.004). The absence of a hernia sack increases the risk for developing shunt-dependent hydrocephalus in patients after prenatal SBA repair. However, as prenatal SBA repair is associated with better motor, urinary, and gastrointestinal function, increased risk of developing shunt-dependent hydrocephalus in fetuses without a hernia sack should not be treated as a contraindication to prenatal intervention.

## 1. Introduction

Spina bifida is the most common defect of the central nervous system, and the second most frequent congenital birth defect, after heart anomalies [1]. The defect develops at the initial stages of the embryonic life (21–28 days of gestation). Spina bifida, which results from abnormal neurulation of the neural tube and failure of the vertebral arches to close over the spinal cord, may occur at any region of the spine, but is most often located at the lumbar region (approximately 62% of the cases), followed by sacral (approximately 20%) or thoracic (approximately 18%) regions [1,2]. Open spinal dysraphism—spina bifida aperta—occurs when the spinal defect and the neural tissue remain exposed and communicate with the outside, whereas closed spinal dysraphism—spina bifida occulta—develops when the neural structures of the bifid spine are covered by skin. The defect may present with a hernia sack—myelomeningocele (protrusion of the neural placode behind the skin surface and expansion of the subarachnoid space)—or without the sack—myeloschisis (the placode remains at the skin level and the spinal canal is directly exposed). The defect is accompanied by Chiari type II malformation of varying severity [3].

Incomplete closure of the spinal neural tube may be visualized on ultrasound as early as at the end of gestational week 10. At 21 weeks of gestation, the defect is diagnosed in 87% of the affected cases [4]. Apart from diagnosing spina bifida, it is also possible to observe on ultrasound the so-called ‘lemon’ and ‘banana’ signs, which accompany secondary hydrocephalus associated with the bifid spine. Nuclear magnetic resonance (NMR) test is recommended if spinal dysraphism in the fetus is suspected. The test allows to confirm or exclude the presence of the defect in the developing fetus. The test will allow for a precise evaluation of the extent of the spinal split, its length and width, the presence or absence of a hernia sack, and the location of the neural structures (the neural plate and the spinal nerves). If a hernia sack is present (myelomeningocele), its size is described in three dimensions, which allows to measure its volume and content. The severity of the Chiari malformation, size of the cerebral ventricles, and the brain structures are evaluated during the test. Moreover, fetal NMR is used to assess the development of the remaining organs and systems to exclude other congenital defects. The diagnosis of spina bifida aperta with hydrocephalus in many cases allows for prenatal surgical closure of the defect. The procedure is performed between 21 and 25 weeks of gestation. Follow-up NMR at 4–6 weeks after surgery is necessary for all prenatal repairs of spina bifida aperta to assess the effectiveness of the intervention. At follow-up, the area of the spinal split is evaluated, together with the surgical site, severity of hydrocephalus, and herniation of the posterior fossa structures.

Apart from the NMR test, an ultrasound evaluation is required every two weeks. Lower risk of developing shunt-dependent hydrocephalus is among the greatest benefits of prenatal repairs of spina bifida aperta. The risk of developing shunt-dependent hydrocephalus in patients undergoing postnatal repair of spina bifida aperta has been estimated at approximately 85% [5].

In our study sample, hydrocephalus was observed in 12% of the patients after prenatal closure of spina bifida aperta. Comparable results have been reported by other centers that perform surgical repairs of spina bifida aperta [5]. Prenatal repair is associated with improved motor function of the lower extremities, as well as bladder and bowel function, as it shortens the exposure of the placode and the spinal nerves to the toxicity of the amniotic fluid and decreases the mechanical trauma to the neural structures resulting from fetal movement within the uterus [6].

## 2. Objectives

The aim of the study was to compare the incidence of shunt-dependent hydrocephalus in patients after prenatal repair of spina bifida aperta versus the presence (myelomeningocele) or absence (myeloschisis) of a hernia sack.

## 3. Material and Methods

A total of 184 prenatal repairs of myelomeningocele using open fetal surgery were performed at our Center between 2005 and 2020. Out of them, 96 patients with spina bifida aperta, located between L2–L3 and S1–S2, and Chiari II, were selected for the study. In all 96 patients, the split started at L2–L3 and ended at S1–S2. The number of vertebral arches that failed to close was the same in all cases. 

The results of the diagnostic MRI, performed between 21 and 24 weeks of gestation, were assessed (Figure 1 and Figure 2). During the test, the following parameters were evaluated: size and location of the spina bifida, presence and size of the hernia sack, size of the cerebral ventricles, structure of the posterior fossa, and severity of the Chiari malformation. Prenatal fetal NMR allowed to evaluate the development of the remaining fetal organs. Only fetuses with no other developmental defects were deemed eligible for the prenatal intervention. The NMR findings were compared to the perioperative imaging of the spinal split and hernia sack. After open fetal prenatal repair of spina bifida aperta, a follow-up ultrasound was performed at 2 weeks postoperatively, and repeated every 10–14 days. A follow-up fetal NMR was performed at 6–8 weeks postoperatively (Figure 1 and Figure 2).

The next follow-up test, a fontanelle ultrasound, was performed every 5–7 days of neonatal life to assess the size of the cerebral ventricles and location of the posterior fossa structures. The dynamics of hydrocephalus progression was assessed based on the measurements of the lateral ventricular size (anterior horns and stem) and increased resistance index of the anterior cerebral artery (RI ACA) on follow-up ultrasound, as well as the measurements of the head circumference and palpable tension of the anterior fontanelle. The measurements were then used to determine the need to place Ventriculo-Peritoneal shunt (V-P-shunt) if the anterior horn and lateral ventricular size exceeded the 90th percentile on the pediatric ultrasound scale, and continued rising, RI ACA was >0.75, and increased tension of the anterior fontanelle failed to resolve [7,8] (Figure 3 and Figure 4).

## 4. Results

A total of 96 patients who had undergone prenatal repair for spina bifida aperta using open fetal surgery were analyzed. The split was located from L2–L3 to S1–S2 in all fetuses. Perioperative assessment of the level of the split and the number of vertebral arches that failed to close was consistent with the results of the earlier NMR evaluation.

The patients were divided into two groups:

Group I—12 (12.5%) patients with no hernia sack on diagnostic imaging and during surgical repair (Figure 1).

Group II—84 (87.5%) patients with a hernia sack on diagnostic imaging and during surgical repair (Figure 2).

Based on the hernia sack volume, the patients from Group II were subdivided as follows:

II A—35 (41.7%) patients with a small hernia sack

(sack volume of 1320–2400 mm^3^) 

II B—40 (47.6%) patients with a medium hernia sack 

(sack volume of 1520–3120 mm^3^) 

II C—9 (10.7%) patients with a large hernia sack 

(sack volume of 3570–12,670 mm^3^) (Table 1).

Statistical analysis was performed using the chi-square test and demonstrated that the absence of a hernia sack in patients with spina bifida aperta is statistically significantly less common as compared with the presence of a sack (*p* > 0.0001). The same test revealed that the incidence of a large hernia sack (>3570 mm^3^) is statistically significantly lower as compared with a medium (*p* > 0.000019) and a small (*p* > 0.000056) hernia sack (Table 1).

At 6–8 weeks after prenatal closure of spina bifida, a control NMR test revealed the stage of Chiari to be unchanged, with gradually enlarging ventricles, in all patients who required V-P shunting. As for patients who did not need shunting, a control NMR at 6–8 weeks after prenatal closure of spina bifida confirmed reversed herniation of the posterior fossa in the foramen magnum, with either unaffected or minimally (max. 2 mm) enlarged ventricles.

The chi-square test also revealed that V-P shunting was statistically significantly less common in patients after prenatal SBA repair (*p* > 0.00001) (Table 2). Out of both groups (I and II), a total of 12 (12.5%) patients required postoperative placement of the V-P shunt owing to progressing hydrocephalus. The shunt was placed between day 21 and 39 of neonatal life. Out of the 12 patients in question, 10 were members of Group I (no hernia sack) and 2 were members of Group II (hernia sack), sub-group IIB (medium hernia sack). Statistical analysis revealed that V-P shunting was statistically significantly more often necessary in Group I (*p* > 0.004) as compared with Group II (*p* > 0.00001) (Table 2).

The positive likelihood ratio (PLR), positive predictive value (PPV), and negative predictive value (NPV) were used for statistical analysis and indicated that the risk for V-P shunting in patients after prenatal repair for spina bifida without a hernia sack was estimated at 87%. In turn, the likelihood that patients after prenatal repair for spina bifida with a hernia sack will require shunting was calculated at 8.3%. Statistical analysis of the sack volume found no statistically significant differences between the size of the sack and the necessity to place a V-P shunt.

## 5. Discussion

Prenatal repair of myelomeningocele has been performed since 1998 [9].

The randomized results of the first MOMS trail, which compared the outcomes of pre- and postnatal MMC repairs, demonstrated that prenatal MMC closure statistically significantly decreased the risk for the development of hydrocephalus and improved the motor function of the lower extremities. Later studies proved that infants after prenatal MMC repair had better bladder and bowel function [10,11].

Currently, fetal MMC repairs are performed at numerous centers in the USA, Europe, and Brazil, both open (small incision is made directly above the spina bifida aperta on the exposed maternal uterus) and fetoscopic (the ports are inserted into the exposed maternal uterus).

Fetoscopic surgery is associated with lower maternal morbidity and increases the chance of natural vaginal delivery. Still, Fetoscopic interventions are not without certain limitations. In the case of extensive myelomeningocele and large exit site, it is not possible to perform an effective and complete reconstruction of the dura mater, muscles, and skin using the endoscopic route.

The absence of a hernia sack usually causes significant skin deficit and greater tethering of the placode and the spinal nerves. Therefore, open fetal surgery for spina bifida aperta is recommended in such cases, which is consistent with our observations. During open fetal surgery, it is possible to cover the exposed spinal cord and spinal nerves with the dura mater, which in the future will significantly lower the risk for secondary tethering of the spinal cord. Open fetal surgery also allows to perform a more extensive and precise transposition of the paravertebral muscles, restoring normal structure, symmetry, and anatomy, which in turn decreases the risk for developing scoliosis.

OFS for hysterotomy uses two methods: one with staples and the other with DeBakey clamps and diode laser beam. In the latter, the fetal membranes are tightly fixed to the uterine wall. Even though the method without the staples is more time-consuming, it allows to make a more precise incision and calculate its length [12].

According to the two-hit hypothesis by Heffez, abnormal neurulation of the spinal cord and its subsequent progressing chemical (due to amniotic fluid toxicity) and mechanical (due to fetal movements in utero) trauma may lead to irreversible damage to the tissues of the central (spinal cord) and peripheral (spinal nerves) nervous systems [6].

Lack of a hernia sack may intensify the damage for two reasons. First, the sack is filled with fluid, which may serve as a hydrostatic cushion and lessen the impact of mechanical trauma. Secondly, the fluid in question is a mixture of the cerebrospinal fluid and the amniotic fluid, which permeated through the thin wall of the sack. As a result, the concentration of the substances in the amniotic fluid that are toxic for the neural structures is significantly lower. The absence of a hernia sack may intensify the mechanical and chemical damage to the placode and the spinal nerves. Progressing mechanical and chemical damage will amplify the inflammatory reaction, which in turn will intensify the adhesion of the neural tissue with the surrounding dura mater, muscle, and skin. Greater inflammatory reaction and adhesion increase the tethering of the cord at a larger area, which lowers the chances for its complete untethering during surgery. Untethering of the spinal cord and nerves during a myelomeningocele repair allows for the spinal cord to be released within the vertebral canal, which in turn allows the herniated posterior fossa structures to move upwards, above the foramen magnum. Release of the posterior fossa enhances the hemodynamic conditions of the cranio-cervical junction, improving the circulation of the cerebrospinal fluid and lowering the risk for the development of shunt-dependent hydrocephalus. Failure to completely untether the spinal cord and nerves during the myelomeningocele repair increases the risk for hydrocephalus and the need for ventriculoperitoneal shunting [3,11].

An inflammatory reaction in the skin and dura mater in fetuses with spina bifida aperta intensifies after 24 weeks of gestation [13]. An intensified inflammatory response in the tissue surrounding the neural structures results in intensified inflammatory response in the placode and spinal nerves, which in consequence increases their tethering. The abovementioned might explain why the risk for developing shunt-dependent hydrocephalus is significantly higher in children who underwent postnatal repair of spina bifida aperta, even with a hernia sack, as compared with prenatal surgery.

When discussing management options, it is essential to inform the parents of the affected child about the prognosis and the complications, and hydrocephalus remains one of the most severe complications associated with spina bifida. The parents and the patients are concerned about the risk associated with another anesthesia and surgical intervention in the child, should shunting prove to be necessary. Another cause for concern is the fact that the shunt system may suffer malfunction, disconnection, or blockage, which will require system replacement [14]. Knowledge of the fact that the absence of the hernia sack in spina bifida aperta statistically significantly increases the risk for developing shunt-dependent hydrocephalus, even despite prenatal repair of the defect, allows to predict the possibility of such complications with more precision. Importantly, prenatal repair of spina bifida aperta, regardless of the presence or absence of a hernia sack, offers a chance at better motor function of the lower extremities and improved neurogenic bladder and bowel function [5,10]. Therefore, elevated risk for shunt-dependent hydrocephalus in patients with spina bifida aperta without a hernia sack should not constitute a contraindication to fetal surgery. Obviously, the parents should be made aware of the risks and benefits to make a fully informed decision about the surgical intervention. If they give their consent for prenatal surgery, they will be better prepared for the possibility of shunt placement, should hydrocephalus develop in their infant. 

## 6. Conclusions

The absence of a hernia sack increases the risk for developing shunt-dependent hydrocephalus in patients after prenatal repair of spina bifida aperta. However, as prenatal closure of spina bifida aperta offers a chance to maintain improved motor function as well as urinary and gastrointestinal tract function, elevated risk for shunt-depended hydrocephalus should not be perceived as a contraindication to prenatal intervention.

## Figures and Tables

**Figure 1 diagnostics-13-00343-f001:**
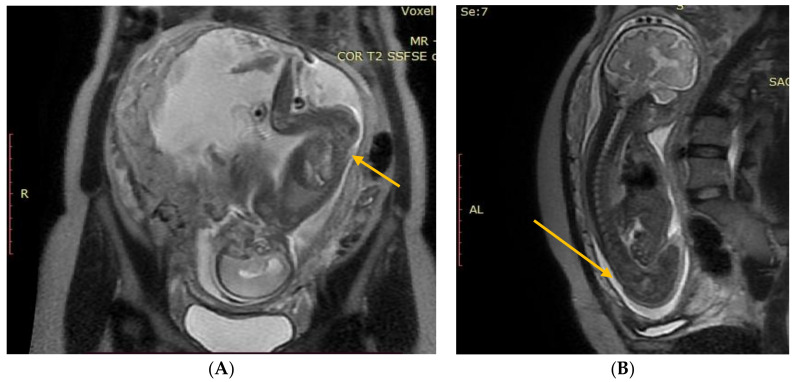
(**A**) Fetal NMR at 21 weeks of gestation. The arrow indicates the myeloschisis, with the area of contact between the split and the uterine wall. (**B**) Fetal NMR at 26 weeks of gestation (4 weeks after intrauterine myeloschisis repair); the arrow indicates the site after a three-layer reconstruction of spina bifida aperta.

**Figure 2 diagnostics-13-00343-f002:**
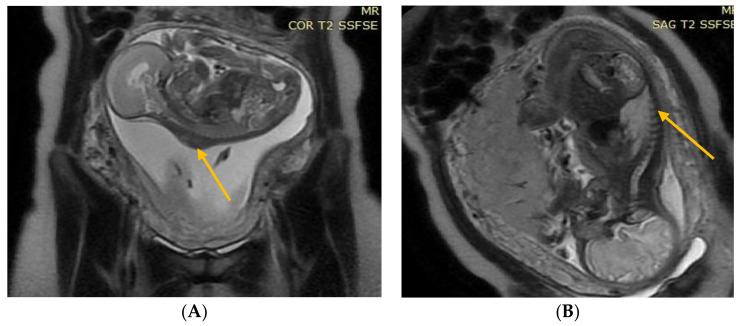
(**A**) Fetal NMR at 22 weeks of gestation. The arrow indicates the myelomeningocele. (**B**) Fetal NMR at 26 weeks of gestation (4 weeks after intrauterine myelomeningocele repair); the arrow indicates the site after a three-layer reconstruction of spina bifida aperta, with the area of contact between the repair site and the uterine wall.

**Figure 3 diagnostics-13-00343-f003:**
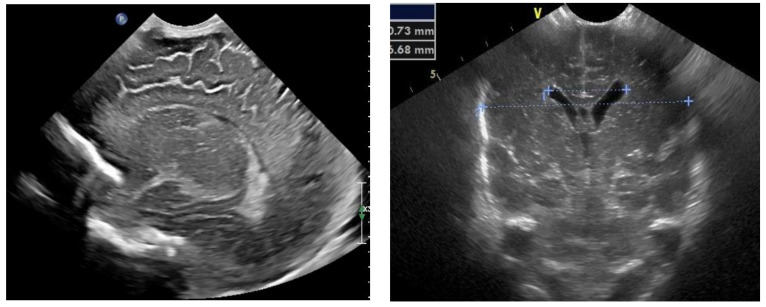
Normal appearance and size of the ventricles on fontanelle ultrasound.

**Figure 4 diagnostics-13-00343-f004:**
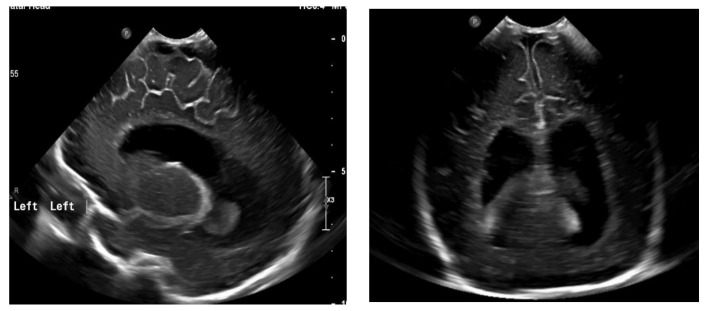
Ventriculomegaly, which requires V-P shunting on fontanelle ultrasound.

**Table 1 diagnostics-13-00343-t001:** The incidence of hernia sack in the study groups and sub-groups.

Group	Sub-Group	Sack Volume [mm^3^]	Number of Patients	Statistical Significance (*p*)
I	-	no sack	12	>0.0001
II	II A small sack	1320–2400	35	>0.000056
	II B medium sack	1520–3120	40	>0.000019
	II C large sack	3570–12,670	9	<0.005

**Table 2 diagnostics-13-00343-t002:** The necessity of placing a V-P shunt in the study groups and sub-groups.

Groups and Sub-Groups	Number of Patients	Need for a V-P Shunt Number of Patients	No Need for a V-P Shunt Number of Patients	Statistical Significance (*p*)
Group I	12	10	2	>0.004
Group IIA	35	-	35	-
Group IIB	40	2	38	>0.00001
Group IIC	9	-	9	-
Total	96	12	84	>0.00001

## Data Availability

The data presented in this study are available upon request from the corresponding author. Data are not publicly available because they are contained in individual cases histories and making them public could identify the patients.

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
