# Peer review of "Absence of a Hernia Sack in Patients Undergoing Prenatal Repair of Spina Bifida Increases the Risk of Developing Shunt-Dependent Hydrocephalus"

_diagnostics, 2023, doi:10.3390/diagnostics13030343_

Round 1

Reviewer 1 Report

This is a good manuscript with important findings. The English writing in the results section needs to be a little bit clearer. This is also true of the abstract as well. These are minor revisions in sentence structure. For example

The chi-square test also revealed that no need of V-P shunting in patients after pre- 146 natal repair for spina bifida aperta is statistically significantly higher than the need to 147 place the shunt (p>0.00001) (Table 2).

. In this study, we demonstrated that patients after prenatal SBA repair statistically sig- 18 nificantly less often required ventriculoperitoneal (V-P) shunting (p>0.00001). Patients who needed 19 the shunt statistically significantly more often belonged to Group I (p>0.004).

These two sentences are a little awkward

Next the authors may want to consider revision on one paragraph in the discussion section

The absence of a hernia sack usually causes significant skin deficit and greater teth- 178 ering of the placode and the spinal nerves. Therefore, open fetal surgery for spina bifida 179 aperta is recommended in such cases, which is consistent with our observations. During 180 open fetal surgery, it is possible to cover the exposed spinal cord and spinal nerves with 181 the dura mater, which in the future will significantly lower the risk for secondary tether- 182 ing of the spinal cord. Open fetal surgery also allows to perform a more extensive and 183 precise transposition of the paravertebral muscles, restoring normal structure, symmetry, 184 and anatomy, which in turn decreases the risk for developing scoliosis.  

Fetoscopic techniques now allow for closure of myeloschisis using a patch and myofascial flap to get layered closure a good seal and promote neoduramater formation. The morbidity to the mother is less using the fetoscopic approach

Overall though this is a well done study with important results. It just needs minor revisions. Thank you to the authors for sharing their work and allowing me to review this fine manuscript.

Author Response

Fetoscopic techniques now allow for closure of myeloschisis using a patch and myofascial flap to get layered closure a good seal and promote neoduramater formation. The morbidity to the mother is less using the fetoscopic approach

We wish to thank the Reviewer for the feedback and invaluable comments. We proofread the manuscript and made all the necessary improvements. We followed the advice of the Reviewer and clarified the pinpointed sections of the text, to improve cohesion and coherence. In the Discussion, we included the comment that maternal morbidity is lower in the fetoscopic approach, and that natural vaginal delivery of the fetus after prenatal SB repair is indeed possible.

Reviewer 2 Report

the authors present a study to evaluate the incidence of the absence of hernia sack  in developing hydrocephalus.

The question is interesting in order to answer the question of the family before birth and also to clinician in the management of these children.

However, I have few questions to the authors. In general I think that the authors did not control there results with the others factors that are known to increase the risk f hydrocephalus

Majors issues:

1) was there a significant difference in the anatomical level of the neural defect between the groups ? Indeed, it well know that he higher the defect the greater the risk of developing a hydrocephalus is. 

2) was there more associated cerebral anomalies in the Group 1  (Chiairi, corpus callosum ...) ?  For the same reasons as above

3) why only 96 patients were included : what were the reasons for the 88 exclusion (184 total patients) ? This should be well explain in the methods section.

Minors:

1) The authors should temper their statement on the effect of the prenatal closure on the bladder and bowel management; Even though, the authors' group published one article on the subject, most of the studies did not find a positive effect. Remove ref 5 in the discussion as the MOMS did not clearly conclude on the subject 

Author Response

We wish to thank the Reviewer for the much appreciated feedback, invaluable comments and suggestions. Our comments and explanations are listed below.

Major issues:

  1. We agree that the incidence of shunt-dependent hydrocephalus depends on the level of the split. That is why the patients in out study population – as stated in the Material and Methods section of the manuscript – had the same level of split.

  2. As stated in the Material and Methods section of the manuscript, all patients presented with the Chiari malformation type II. No more coexisting cerebral abnormalities were analyzed as they may have a very different and a very complex effect on the hydrocephalus than Chiari II. However, we intend to conduct such an analysis in the future, comparing pre- and postnatally operated patients.

  3. As abovementioned, we included only patients with the same level split (L2-L3 to S1-S2) and with Chiari II. Between 2005 and 2020, only 96 patients met the two criteria. The location of the split in the remaining patients who underwent prenatal repair for spina bifida at that time was different so we excluded them – as the Reviewer noticed in point 1, different levels of split are associated with the different risks for shunt-dependent hydrocephalus.

Minor issues:

We agree with the Reviewer’s comment that MOMS did not clearly conclude on the effect of prenatal SB repairs on the bladder and bowel functions, so we removed the reference from the place in question. A center in Zurich had similar observations to our findings on the effect of SB repair on the function of the urinary bladder and we included their study into the Reference section.

Round 2

Reviewer 2 Report

I thank the authors for their reply.

However, some points still need to be clarified

1) there is a big difference between a L2L3 and a S1S2 level in term of hydrocephalus incidence. I did well read that all included patients have a level between L2 and S2 but my question was : within L2 and S2 what was the repartition of the anatomical level of the MMC between the groups ? were there more patients with high defect (above L4) in one group ?

2) the authors reply is not acceptable in this form. In their study, in order to conclude what they concluded they need to, at least, looked at the rate of post natal Chiari. Indeed, reversibility of the initial Chiari can happened after prenatal closure. The presence or not of a Chiari at the time of the shunt is an information absolutely compulsary to include in the results section and in the statistical analysis. Indeed, if the rate of Chiari is significantly higher in the group 1, the conclusion that the size of the sack is responsible of a higher shunt rate implantation might be questionable.

3) 

Author Response

I thank the authors for their reply.

However, some points still need to be clarified

1) there is a significant difference between a L2L3 and a S1S2 level in term of hydrocephalus incidence. I did well read that all included patients have a level between L2 and S2 but my question was : within L2 and S2 what was the repartition of the anatomical level of the MMC between the groups ? were there more patients with high defect (above L4) in one group ?

2) the authors reply is not acceptable in this form. In their study, to conclude what they concluded they need to, at least, looked at the rate of post natal Chiari. Indeed, reversibility of the initial Chiari can happen after prenatal closure. The presence or not of a Chiari at the time of the shunt is an information absolutely compulsory to include in the results section and in the statistical analysis. Indeed, if the rate of Chiari is significantly higher in the group 1, the conclusion that the size of the sack is responsible of a higher shunt rate implantation might be questionable.

Dear Reviewer,

thank you for your feedback and invaluable comments.

  • We were indeed not clear enough about the level of the split. In all patients from both groups, the split started at L2-L3 and ended at S1-S2, so the ‘porta’ of the split and the number of vertebral arches which failed to close was the same in all cases.

Patients who did not meet the criteria for the split parameters (the split started or ended at various levels than our study population) were excluded from the analysis. 

As per your suggestion, more detailed explanation has been included in the text.

  • We are grateful for this comment, it has been most helpful.

We should have included the information that in all patients who required V-P shunting, the postnatal Chiari - evaluated on control NMR at 6-8 weeks after prenatal closure of spina bifida - remined unchanged and the ventricles were gradually enlarging. As far as the patients who had no need of shunting were concerned, control NMR at 6-8 weeks after prenatal SB closure revealed resolution of the herniation in the foramen magnum and not enlarged ventricles.   

The changes have been made and included in the text.   

Round 3

Reviewer 2 Report

Manuscript changes are sufficient for publication

Author Response

Dear Reviewer,

thank You for your help and cooperation in improving the manuscript.